# Early Alterations in Structural and Functional Properties in the Neuromuscular Junctions of Mutant FUS Mice

**DOI:** 10.3390/ijms24109022

**Published:** 2023-05-19

**Authors:** Marat A. Mukhamedyarov, Aydar N. Khabibrakhmanov, Venera F. Khuzakhmetova, Arthur R. Giniatullin, Guzalia F. Zakirjanova, Nikita V. Zhilyakov, Kamilla A. Mukhutdinova, Dmitry V. Samigullin, Pavel N. Grigoryev, Andrey V. Zakharov, Andrey L. Zefirov, Alexey M. Petrov

**Affiliations:** 1Department of Normal Physiology, Kazan State Medial University, 49 Butlerova St., Kazan 420012, Russia; marat.muhamedyarov@kazangmu.ru (M.A.M.);; 2Kazan Institute of Biochemistry and Biophysics, Federal Research Center ‘‘Kazan Scientific Center of RAS”, 2/31 Lobachevsky St., P.O. Box 30, Kazan 420111, Russianikitozz_07@mail.ru (N.V.Z.);; 3Department of Radiophotonics and Microwave Technologies, Kazan National Research Technical University, 10 K. Marx St., Kazan 420111, Russia; 4Laboratory of Neurobiology, Kazan Federal University, Kazan 420008, Russia

**Keywords:** amyotrophic lateral sclerosis (ALS), asynchronous neurotransmitter release, calcium transient, end plate, endocytosis, exocytosis, FUS, neuromuscular junction (NMJ), nicotinic acetylcholine receptor (nAChR), synaptic vesicle, synapsin

## Abstract

Amyotrophic lateral sclerosis (ALS) is manifested as skeletal muscle denervation, loss of motor neurons and finally severe respiratory failure. Mutations of RNA-binding protein FUS are one of the common genetic reasons of ALS accompanied by a ‘dying back’ type of degeneration. Using fluorescent approaches and microelectrode recordings, the early structural and functional alterations in diaphragm neuromuscular junctions (NMJs) were studied in mutant FUS mice at the pre-onset stage. Lipid peroxidation and decreased staining with a lipid raft marker were found in the mutant mice. Despite the preservation of the end-plate structure, immunolabeling revealed an increase in levels of presynaptic proteins, SNAP-25 and synapsin 1. The latter can restrain Ca^2+^-dependent synaptic vesicle mobilization. Indeed, neurotransmitter release upon intense nerve stimulation and its recovery after tetanus and compensatory synaptic vesicle endocytosis were markedly depressed in FUS mice. There was a trend to attenuation of axonal [Ca^2+^]_in_ increase upon nerve stimulation at 20 Hz. However, no changes in neurotransmitter release and the intraterminal Ca^2+^ transient in response to low frequency stimulation or in quantal content and the synchrony of neurotransmitter release at low levels of external Ca^2+^ were detected. At a later stage, shrinking and fragmentation of end plates together with a decrease in presynaptic protein expression and disturbance of the neurotransmitter release timing occurred. Overall, suppression of synaptic vesicle exo–endocytosis upon intense activity probably due to alterations in membrane properties, synapsin 1 levels and Ca^2+^ kinetics could be an early sign of nascent NMJ pathology, which leads to neuromuscular contact disorganization.

## 1. Introduction

Amyotrophic lateral sclerosis (ALS) is a fatal neurodegenerative disease (2–4 per 100,000) affecting both upper (cortical) and lower (spinal) motor neurons. Despite the variability of inherent (~10%) and sporadic (~90%) forms of ALS, a loss of the neuromuscular junction (NMJ) is an early and critical hallmark in both mouse models and ALS patients [1,2,3]. A main hypothesis is that distal axonopathy occurs at the NMJs at early stages before motor neuron degeneration and even clinical manifestation [2,4,5,6,7,8,9]. Accordingly, changes in processes underlying neuromuscular transmission can trigger events in the following denervation, muscle atrophy and motor neuron loss.

Mutations in RNA-binding protein FUS (fused in sarcoma) are the common reason for familiar ALS cases (~4–6%) and were found in a cohort (~0.7–1.8%) of patients with the sporadic form of ALS. FUS mutations result in ~1/3 of juvenile ALS cases [10,11]. Over fifty missense FUS mutations, mostly located in the nuclear localization signal encoding region, are known to provoke the translocation of a significant portion of FUS from the nucleus to the cytoplasm of motor neurons, which seems to be a crucial event in ALS pathogenesis [5,11,12]. Beyond ALS, inclusions containing FUS were detected in the neuronal cytoplasm of ~10% of individuals with frontotemporal dementia characterized by early synaptic loss in brain regions controlling cognition, language and behavior [5,13]. Typical pathological alterations induced by FUS mislocalization include defects in protein synthesis, calcium homeostasis, cytoskeleton dynamics and mitochondrial function [4,5,11,14,15].

Under physiological conditions, neuronal FUS is involved in multiple cellular processes via the regulation of RNA metabolism (splicing, trafficking and translation) in soma and locally in synaptic regions [11,12,16]. Importantly, FUS is abundant in presynaptic nerve terminals of NMJs, indicating its important role in the peripheral synapses [4]. Additionally, FUS is present in the subsynaptic nuclei of the postsynaptic region of NMJs [6]. Hence, FUS could be essential for the maintenance of NMJs.

It is well-known that quickly fatigable motor units and muscles (e.g., medial gastrocnemius, tibialis anterior and extensor digitorum longus) are the most vulnerable and degenerated first in ALS [3], but paralysis of the diaphragm muscle is the cause of lethality. Furthermore, in ~3–5% of cases, the onset of ALS occurs in the respiratory muscles [17]. Note that diaphragm is a highly active and densely innervated muscle with mixed fiber composition [18]. Mouse diaphragms include type I (slow; 11%), IIa (fatigue-resistant, 51%) and IIx (quickly fatigable, 38%) muscle fibers controlled by NMJs with similar morphology [19]. Early synaptic changes in diaphragm muscle were observed in SOD1(G93A) and SOD1(G37R) model mice with ALS [20,21,22,23,24].

In the present study, using fluorescent and electrophysiological approaches, the characterization of early structural and functional alterations of diaphragm NMJs was performed on ΔFUS(1-359) mice at the pre-onset stage. These model mice expressed the human truncated FUS 1–359 protein which lacks RNA-binding capacity and is predominantly cytoplasmic and highly aggregate-prone [25]. This model allows mainly a revelation of the pathological role of FUS aggregation rather than the dysregulation of FUS RNA targets due to the expression of mutant full-length FUS [25,26]. ΔFUS(1-359) mice display the key hallmarks of ALS including oxidative stress and elevated levels of pro-inflammatory cytokines in the spinal cord, FUS-positive inclusions in motor neurons, microglial activation, limb paralysis, muscle waste and denervation, loss of spinal motor neurons, abrupt disease onset and fast progression [27,28,29,30,31]. Transgenic ΔFUS(1-359) mice have no pathological changes or any motor symptoms until 10 weeks of age (pre-onset stage) and fast disease progression occurs during 10–18 weeks of age with a severe motor phenotype [25,26,27,31,32].

Herein, we found an increase in presynaptic protein levels (synapsin 1 and SNAP-25), a suppression of evoked neurotransmitter release and synaptic vesicle endocytosis upon intense activity as well as changes in synaptic membrane properties accompanied by lipid peroxidation at the early stage (6–8 weeks old) in ΔFUS(1-359) mice before motor symptom manifestation. At the onset stage (18–20 weeks old), shrinking and fragmentation of end plates together with a marked decrease in presynaptic protein expression and disturbance of the neurotransmitter release timing occurred.

## 2. Results

### 2.1. Structural Characterization of NMJs: Analysis of End-Plate Fragmentation and Immunolabeling of Presynaptic Proteins

The labeling of nicotinic acetylcholine receptors (nAChRs) with fluorescent α-Btx showed no marked changes in postsynaptic receptor density or their distribution at the pre-onset stage, i.e., 6–8 weeks (Figure 1A,B). The number of end-plate fragments and the intensity of αBtx fluorescence were similar in the diaphragm muscles of FUS and WT mice (Figure 1B). There was also no difference in the area of end plates (Table 1; Appendix A).

The immunostaining of presynaptic proteins revealed the absence in changes in area of synapsin 1, SNAP-25 and synaptophysin-positive regions in FUS vs. WT mice (Table 1). However, immunoexpression of synapsin 1 and SNAP-25 was ~25% higher in FUS mice than it was in WT mice (Figure 1A,C). At the same time, synaptophysin fluorescence was not altered (Appendix A). An estimation of colocalization between presynaptic (synapsin 1 and SNAP-25) and postsynaptic (nAChR) proteins did not reveal significant alterations in FUS vs. WT mice (Table 1).

Thus, the general structure of NMJs, the matching of presynaptic and postsynaptic components as well as the expression of postsynaptic receptors were unaffected in FUS mice at the pre-onset stage. Only an increase in the immunoexpression of presynaptic proteins, synapsin-1 and SNAP-25 (but not synaptophysin) occurred. The presynaptic protein upregulation observed could be a reflection of the compensatory response to dysfunction of synaptic vesicle exocytosis and endocytosis or the stabilization of presynaptic protein RNA. 

At the onset stage (18–20 weeks), fragmentation of end plates and decrease in their areas (without changes in αBtx fluorescence intensity) were observed (Figure 1D,E; Table 1; Appendix A). The immunoexpression of synapsin 1, SNAP-25 and synaptophysin was reduced by ~20–30% (Figure 1D,F; Appendix A). In addition, a decrease in the percentage of the synapsin 1 fluorescence area covered by αBtx fluorescence (M2-coeficient) occurred in FUS mice at the onset stage (Table 1). Hence, the signs of presynaptic and postsynaptic disorganization probably due to the beginning of denervation appear in the diaphragm muscle at the disease manifestation stage. 

### 2.2. Structural Characterization of NMJs: The Changes in Synaptic Membrane Properties at Pre-Onset Stage

Previous studies on mutant superoxide dismutase (SOD1) mice revealed changes in lipid rafts and signs of ceramide metabolism dysregulation in both the spinal cord and NMJs [21,33,34,35]. Furthermore, increased expression of lipid raft scaffold protein caveolin-1 as well as inhibition of glucocerebrosidase GBA2 (an enzyme involved in ceramide production) were beneficial in mSOD animals [35,36,37].

In these experiments, αBtx was used as a marker of the NMJ region in ex vivo nerve-diaphragm preparations (without fixation) and its fluorescence was unchanged in FUS mice at the pre-onset stage (Figure 2A,B). At the same time, labeling with CTxB (a marker of the GM1 ganglioside cluster within lipid rafts) was decreased by ~46% at the junctional regions of FUS compared to that in WT mice (Figure 2A,C). There were no changes in CTxB fluorescence at the extrajunctional membranes (Figure 2C). This suggests a decrease in GM1 ganglioside ordering in the synaptic zones before the downregulation of the presynaptic proteins and alteration of NMJ structure.

Local changes in levels of endogenous ceramide, a product of sphingolipid hydrolysis, can affect lipid packing [38,39]. The incorporation of exogenous fluorescent ceramide is dependent on levels of endogenous ceramide [40]. Staining of plasmalemma with fluorescent ceramide revealed that the intensity of fluorescence reduced by ~30% selectively in the junctional regions of FUS mice (Figure 2D,E). Hence, the synaptic membranes of the FUS have decreased ability to absorb exogenous ceramide.

Alterations in lipid packing and ceramide content can induce oxidative damage or be a result of lipid peroxidation. Using a ratiometric probe, the BODIPY™ 581/591 C11 reagent, a sign of lipid oxidation was found in the synaptic membranes of FUS mice. Indeed, the ratio of green/red fluorescence of the probe was ~23% higher in FUS vs. WT animals (Figure 2F). Thus, disturbances in membrane packing and ceramide metabolism as well as lipid peroxidation can be early features of a nascent pathological process in the NMJ membranes of FUS mice at the pre-onset stage.

### 2.3. Functional Characterization of NMJs in the FUS Mice: Neurotransmitter Reception and Release under Low Calcium in Extracellular Solution

#### 2.3.1. Spontaneous and Evoked Postsynaptic Responses

Postsynaptic electrical responses were recorded using a extracellular microelectrode with 0.5 mM Ca^2+^ and 5 mM Mg^2+^ in a physiological solution [41,42,43]. Spontaneous neurotransmitter release causes transient postsynaptic depolarization in a form of miniature end-plate potential (MEPP). MEPP parameters (rise and decay time and amplitude) were similar in diaphragm NMJs of WT mice and FUS mice at the pre-onset and onset stages (Figure 3A,B). In addition, the frequency of MEPPs was not altered in FUS mice at both stages (Figure 3A,B). Hence, under low [Ca^2+^]_out_, there were no alterations in both the intensity of spontaneous exocytosis and that of currents via nAChR upon a single quantum in the model mice. 

Under low [Ca^2+^]_out_, presynaptic action potential causes uni-quantal EPP and quantal content (m) is assessed as the ratio of the number of stimuli accompanied by EPP to the total number of stimuli [42,44,45]. This ratio (m) correlates inversely with the failure rate of EPPs. The motor nerve was stimulated at low (0.5 Hz) and higher (10 Hz) frequencies (Figure 3C,D). Increases in the frequency of stimulation markedly increased quantal content in the NMJs of both FUS and WT mice. At the pre-onset stage, there were no significant changes in the rise and decay times of EPPs or in quantal content in the NMJs of FUS mice (Figure 3C). A significant increase in quantal content at 0.5 and 10 Hz stimulation was observed in the NMJs of FUS mice at the onset stage. In addition, a decrease in the decay time of EPPs occurred under 10 Hz stimulation in the NMJs of the symptomatic FUS mice (Figure 3D). Hence, an increase in the probability of evoked exocytosis as well as a slight decrease in the duration of the ACh-induced current occurred in the NMJs of FUS mice at the onset stage. 

#### 2.3.2. Estimation of the Real Synaptic Delay and the Kinetics of Neurotransmitter Release

Presynaptic action potential leads to a release of neurotransmitter molecules, which diffuse through the synaptic cleft to postsynaptic receptors, whose activation generates postsynaptic currents (Figure 4). The changes in synaptic delay, a period from the peak of the presynaptic action potential to the onset of EPP, are mainly dependent on the functional state of voltage-gated Ca^2+^ channels and exocytotic machinery.

Minimal synaptic delay (defined as the shortest, 5%, of all synaptic delays) were not changed in the diaphragm muscles of FUS mice at the pre-onset and onset stages (Figure 4A,B). Latency, as the period between the stimulus and presynaptic action potential, was similar in FUS and WT mice (Appendix A). Accordingly, there were no changes in action potential propagation along the motor nerves or in the delay between action potential arrival into the motor nerve terminal and synaptic vesicle fusion in the NMJs of FUS vs. WT mice. 

Analysis of the distribution of synaptic delays allowed a quantification of the timing (synchronicity) of neurotransmitter release, i.e., the coupling of exocytotic events with presynaptic action potential (Figure 4C,D). Exocytotic events can be divided into synchronous and asynchronous release with short (0–8 ms) and longer (8–50 ms) synaptic delays, respectively. Furthermore, synchronous exocytosis mainly occurs within 0–3 ms (the early synchronous component) followed by a late synchronous component (3–8 ms) [43,46]. At the pre-onset stage, the timing of neurotransmitter release (the contribution of early and later synchronous and asynchronous components) was unchanged (Figure 4C). The early synchronous component was more markedly (on ~30–35%) expressed in the NMJs of FUS mice at the onset stage. This was accompanied by a decrease in the late synchronous (by 50%) and asynchronous (by 40%) components of neurotransmitter release in symptomatic FUS vs. WT mice (Figure 4D). Thus, a synchronization of neurotransmitter release with axon action potential occurred in diaphragm NMJs in FUS mice at the onset stage. Together with an increase in quantal content, synchronization can be a compensatory reaction to counteract a failure of neuromuscular transmission in the diaphragm, especially in the conditions of decreased excitability.

### 2.4. Functional Characterization of NMJs in the FUS Mice: Neurotransmitter Reception and Release under Normal External Calcium in Extracellular Solution

Neuromuscular transmission under normal external Ca^2+^ levels relies on a release of several dozen up to a hundred acetylcholine quanta from a single NMJ [47]. Accordingly, evoked synaptic vesicle exocytosis occurs simultaneously in many active zones under normal extracellular Ca^2+^ levels. During prolonged and rhythmic activities, mobilization and endocytotic recycling of synaptic vesicles greatly contribute to the maintenance of neurotransmission. In addition, Ca^2+^ influx through nAChRs in NMJs can affect the excitability of the muscle fiber membrane [48]. In the following experiments, to reveal early neuromuscular transmission alterations, the postsynaptic currents were recorded using the voltage clamp technique with 2.0 mM [Ca^2+^]_out_ in diaphragm NMJs of pre-onset FUS mice (Figure 5).

#### 2.4.1. Spontaneous and Evoked Neurotransmitter Release

Microelectrode recording of MEPCs and EPCs (under 0.05 Hz stimulation) at the clamped membrane potential showed changes in the postsynaptic level in the NMJs of FUS mice at the pre-onset stage (Figure 5A,B). Indeed, a reduction in rise and decay times as well as in the amplitude of MEPCs was found in the NMJs of FUS mice (Figure 5A). At the same time, the frequency of MEPCs was unchanged. This suggests no changes in the intensity of spontaneous release and a decrease in nAChR sensitivity to neurotransmitters under normal calcium levels in the external solution. As a result, a decrease in the amplitude of EPCs in combination with unchanged EPC quantal content was observed in the NMJs of FUS mice. There were no alterations in the rise and decay times of EPCs in the model mice. Thus, under normal extracellular Ca^2+^ levels, the number of quanta released per nerve stimulus did not change, but the postsynaptic membrane sensitivity to acetylcholine was reduced in the NMJs of FUS mice at the pre-onset stage.

#### 2.4.2. Short-Term Plasticity and the Kinetics of Neurotransmitter Release upon Intense Nerve Stimulation

Paired-pulse facilitation is a form of short-term synaptic plasticity, which is manifested as an increase in the amplitude of the second postsynaptic response of two excitatory postsynaptic potentials divided by a short inter-impulse interval [49]. At the inter-impulse interval of 50 ms, paired-pulse facilitation was expressed in the NMJs of FUS mice but not in those of WT mice (Figure 5C). This can reflect changes in residual Ca^2+^ levels after the first exocytotic event, the sensitivity of exocytotic machinery to Ca^2+^ or the initial probability of exocytosis.

Higher-frequency nerve stimulation evoked massive synaptic vesicle exocytosis and, hence, the efficacy of neurotransmission under intense activity relies on the mobilization and endocytotic recycling of synaptic vesicles. In the control, nerve stimulation at 20 Hz led to a depression in the EPC amplitude (Figure 5D), which decreased by ~35% after 180 s of the activity. After the end of the 20 Hz stimulus train, the EPC amplitude was fully recovered within close to one minute (Figure 5D). In the NMJs of the FUS mice, 20 Hz stimulation caused a faster decrease in the EPC amplitude (of ~58% after 3 min of the stimulation) (Figure 5D). As a result, the cumulative amplitude of EPCs by the third min of 20 Hz activity was ~25% lower in the NMJs of FUS mice vs. those of WT mice (Figure 5D). Additionally, the recovery of the EPC amplitude after the 20 Hz stimulus train was suppressed (Figure 5D). Accordingly, the NMJs of the FUS mice were characterized by a reduction in provoked neurotransmitter release under intense activity as well as a disturbance of neurotransmitter release recovery after tetanus.

### 2.5. Functional Characterization of NMJs in the FUS Mice: Estimation of Synaptic Vesicle Endocytosis

Decreased neurotransmitter release during intense activity can be accompanied by a reduction in compensatory synaptic vesicle endocytosis. Labeling of synaptic vesicles with FM1-43 dye in diaphragm muscles showed a marked decrease (of ~43%) in the dye uptake in response to the induction of massive exo–endocytosis in the motor nerve terminals of the FUS mice at the pre-onset stage (Figure 6). Furthermore, analysis of FM1-43 fluorescence intensity in individual NMJs revealed an increase in the portion of nerve terminals with reduced FM 1-43 loading in the FUS mice vs. that in the WT mice. The analyzed area of nerve terminal fluorescence was similar between the NMJs of FUS and WT mice (98 ± 62 µm^2^; 176 NMJs from 4 mice vs. 95 ± 51 µm^2^, 176 NMJs from 4 mice). Hence, a reduction in exo–endocytosis upon intense activity can be an early pathological sign in the NMJs in FUS mice. 

### 2.6. Functional Characterization of NMJs in the FUS Mice: Intraterminal Ca^2+^ Dynamics upon Nerve Stimulation at Different Frequencies

Calcium dynamics in the nerve terminals of the levator auris longus were estimated using Oregon Green 488 BAPTA-1 and Magnesium Green indicators with high and lower affinity to Ca^2+^, respectively (Figure 7). The former was used to estimate the calcium transient in response to low-frequency stimulation (0.5 Hz) of the motor nerve; the latter was used to record intraterminal calcium dynamics upon stimulation at higher frequencies (10, 20 and 50 Hz). The initial calcium fluorescence was similar between the nerve terminals of FUS and WT animals. The peak amplitude as well as temporal parameters of the calcium transient were the same in the motor nerve terminals of FUS and WT mice (Figure 7A). 

Higher-frequency stimulus trains at 10, 20 and 50 Hz led to a fast increase in dye fluorescence following the plateau phase or a slow growth in fluorescence with the time of the stimulation (Figure 7B). The maximal amplitudes of the calcium responses were similar in the nerve terminals of FUS and WT mice at all frequencies tested. There was only a trend of decrease in the peak amplitude at 20 Hz stimulation in the nerve terminals of FUS mice at the pre-onset (*p* = 0.074) and onset (*p* = 0.097) stages. The difference in amplitude reached a statistically significant level (*p* = 0.039) if presymptomatic and symptomatic mice were combined. Assessment of the area under the curves of calcium dynamics (as an integral indicator of intraterminal calcium increase) showed a decrease (*p* = 0.051) in cytosolic calcium under 20 Hz stimulation in the nerve terminals of the FUS mice at the pre-onset stage (Appendix A). Hence, the intraterminal calcium transient upon single stimuli and calcium dynamics during intense activities were not markedly altered at the pre-onset and onset stages of the model pathology. Only a trend of reduction in the cytosolic calcium concentration was observed under 20 Hz activity. 

## 3. Discussion

The main findings of the present study are as follows: membrane abnormalities (lipid peroxidation and disordering) and upregulation of presynaptic proteins (SNAP-25 and synapsin 1) are early structural signs of a nascent pathology of diaphragm NMJs; a decrease in both synaptic vesicle exocytosis and compensatory endocytosis upon intense motor nerve activity as well as a deficiency in the recovery of neurotransmitter release after tetanus were present in the NMJs of ΔFUS(1-359) mice at the pre-onset stage (Figure 8). At the same time, the end-plate morphology, neurotransmitter release in response to low-frequency stimulation under both normal and lowered external Ca^2+^ levels, the synchronicity of exocytotic events and intraterminal Ca^2+^ dynamics during low- and high-frequency axonal activity were not markedly affected at the pre-onset stage. There was only a trend of decrease in axonal Ca^2+^ elevation under 20 Hz nerve stimulation. However, end-plate shrinking and fragmentation, a decrease in synapsin 1, SNAP-25 and synaptophysin levels, and a synchronization of exocytotic events occurred in the NMJs of ΔFUS(1-359) mice at the onset stage. Probably, these late alterations could be consequences of the beginning of muscle denervation. 

There were the changes in postsynaptic structure and neurotransmitter reception in the NMJs of mutant FUS mice. The formation of postsynaptic pretzel-like structures is mainly dependent on the expression of specific genes in subsynaptic nuclei, the small GTPase-dependent membrane transport of nAChRs, and cytoskeleton and scaffold proteins [50,51]. FUS was identified in the postsynaptic regions of central glutamatergic synapses as a component of the RNA–protein complex associated with NMDA-receptors [12,16]. In a knock-in FUS-ALS mouse model, expressing a truncated FUS protein with a lack of 20 C-terminal amino acids, the area and total number of end plates in the hind limb muscles were reduced in newborn. These postsynaptic defects are attributable to the toxicity of the cytoplasmic FUS protein in muscle fibers and the decreased ability of normal FUS located in subsynaptic nuclei to increase the expression of genes (e.g., nAChR subunits) important for NMJ stability [6]. However, there were no marked changes in end-plate morphology in the ΔFUS(1-359) mice at the pre-onset stage; a fragmentation of end plates and decrease in their areas were observed. This can reflect the disruptive action of mutant FUS in diaphragm muscle fibers at a later stage. Interestingly, that extracellular vesicles derived from ALS myotubes contain FUS and its binding partners, and these vesicles can induce damages in motor neurons [52].

The amplitude, rise and decay times of miniature end-plate responses were slightly decreased under normal (but not low) external Ca^2+^ levels in the NMJs of ΔFUS(1-359) mice at the pre-onset stage. This indicates the Ca^2+^-dependent partial inactivation of postsynaptic nAChRs as a potential early event in the ALS model. In this scenario, the influx of Ca^2+^ via nAChRs can negatively modulate their activity or external Ca^2+^ can act on nAChRs indirectly, e.g., via Ca^2+^-sensing receptors [48,53]. Accordingly, a decrease in external Ca^2+^ concentration could remove the inhibition of nAChRs in ΔFUS(1-359) mice. Note that Ca^2+^ influx through nAChR can activate inositol 1,4,5-trisphosphate receptors enriched in the postsynaptic compartment surrounding the subsynaptic nuclei, and this calcium signaling is crucial to the stabilization of NMJs [48,54]. Hence, the suppression of currents via nAChRs at the pre-onset stage might be a reason for NMJ disorganization in ΔFUS(1-359) mice at the later stage. Another postsynaptic change in the form of an increase in the mean amplitude of miniature end-plate responses and decrease in their rise times were found in the NMJs of SOD1(G93A) mice at the presymptomatic phases of the disease [20,21]. This suggests that partially different alterations can occur in various ALS models. 

There were abnormalities of presynaptic organization and function in the NMJs of FUS mice. Using super-resolution imaging, FUS was localized in the central synapses predominantly near synaptic vesicles of the reserve pool at the presynaptic sites, and FUS accumulation at an early disease stage can lead to the stabilization of synaptic RNA, probably contributing to synaptopathy [55]. Presynaptic localization of FUS was found in NMJs in non-transgenic mice, and overexpression of human wild-type FUS finally caused a marked loss of NMJs and the synaptic vesicle pool as well as mitochondrial vacuolation [4]. Similarly, in FUSP525L and FUSR521C mutant mice, a reduction in synaptic vesicle density occurred at the early stage [7]. Accordingly, increased synapsin 1 and SNAP-25 expression in the NMJs of ΔFUS(1-359) can reflect the stabilization of local RNA at the pre-onset stage. In turn, upregulation of synapsin 1, which retains synaptic vesicles in a non-active cluster [56], can explain the suppression of both exocytotic neurotransmitter release and compensatory endocytosis upon intense activity in the NMJs of ΔFUS(1-359) mice. Alternatively, the depression in exo–endocytosis could be a result of altered intraterminal Ca^2+^ dynamics during high-frequency nerve activity, but the elevation of cytosolic Ca^2+^ was only marginally decreased under 20 Hz activity and unchanged under 50 Hz nerve stimulation. Surprisingly, intraterminal Ca^2+^ transients upon a single stimulus and [Ca^2+^]_in_ kinetics during tetanus at different frequencies were not markedly affected in the motor nerve terminals of ΔFUS(1-359) mice even at the onset stage. 

Disturbance of synaptic vesicle endocytosis can be sign of nascent NMJ pathology. Neuronal expression of the FUS501 proteins in *Caenorhabditis elegans* resulted in an emergence of a population of large endosome-like structures at the NMJs [57]. Using a *Drosophila* model of ALS, it was revealed that TDP-43 toxicity results in part from impaired activity of the synaptic CSP/Hsc70 chaperone complex impacting the function of endocytotic protein dynamin at the NMJs [58]. 

The expression of SNAP-25 affects short-term plasticity, i.e., paired-pulse facilitation [59,60]. Increased expression of SNAP-25 might contribute to increase paired-pulse facilitation under a 50 ms inter-pulse interval at the pre-onset stage. Probably, an increased probability of neurotransmitter release can counteract to some extent the more expressed suppression of neurotransmission under intense activity. Indeed, under the condition of low-frequency stimulation (at normal external Ca^2+^) and decreased presynaptic excitability (low [Ca^2+^]_out_, high [Mg^2+^]_out_) synaptic vesicle exocytosis and its timing were not changed in the ΔFUS(1-359) mice at the pre-onset stage. This points to the unchanged Ca^2+^ sensitivity of the vesicle fusion machinery. Noticeably, an increase in the quantal content of EPPs upon low-frequency simulation was observed at the pre-symptomatic stage in the high-copy-number (SOD1G93A)1Gur strain [20], but not in the low-expressing (SOD1-G93A)^dl^1Gur strain [21].

Only at a later stage, low [Ca^2+^]_out_ quantal content as well as a low portion of synchronous exocytotic events were elevated despite the decreased expression of SNAP-25 and synaptophysin at the NMJs of ΔFUS(1-359) mice. Such synchronization of neurotransmitter quantal release can be a compensatory reaction to alleviate a failure of neurotransmission [46]. One possibility is that decreased synapsin expression can synchronize neurotransmitter release and increase the release probability or size of the readily releasable pool [61,62,63]. Interestingly, synchronization of neurotransmitter exocytotic events can occur in response to the activation of K_ATP_ channels [41], which can be stimulated via ATP depletion in neurodegenerative diseases [64]. Defective ATP production is highly possible in the NMJs of FUS model mice. Indeed, FUS participates in the anchoring of mitochondria into motor nerve terminals via an interaction with syntaphilin, and FUS mutations affect the number and mobility of mitochondria in the synapses [14]. Prior to motor neuron loss, FUSP525L and FUSR521C mutant mice demonstrated an early axon retraction in the tibialis anterior, which was accompanied by a reduction in the number of normal mitochondria without dilation, vacuolation and crista disorganization [7]. In ΔFUS(1-359) mice, ATP production and mitochondria functioning have not been tested yet. Of course, there are other mechanisms underlying presynaptic pathology in ALS, e.g., alterations in axonal transport, active zone organization, local protein synthesis, and glial cell-driven inflammation [2,15,65,66]. 

Alterations in synaptic lipids occur in the NMJs of FUS mice; lipid metabolism is altered in patients with ALS, and an accumulation of “toxic” lipids (ceramide, lysophosphatidylcholine, some oxysterols) can contribute to motor neuron death and NMJ damages [67,68,69,70,71,72]. The levels of sphingosine were markedly increased in the spinal cord of ΔFUS(1-359) mice at the terminal stage of the disease [73]. Studies on hSOD1G93A and Sigma receptor 1 knockdown cells as well as SOD1G93A mice suggest a contribution of lipid peroxidation, lipid raft disruption and ceramide accumulation to early pathogenesis in ALS [21,74,75,76]. Indeed, decreased staining of NMJ membranes with the lipid raft marker, lipid peroxidation as well as the decreased ability of the synaptic membrane to bind fluorescent ceramide were observed in ΔFUS(1-359) mice at the pre-onset stage. The latter can be a result of the accumulation of endogenous ceramide in the NMJs [40]. Lipid raft disruption and ceramide accumulation occur in response to even short (6–12 h) skeletal muscle disuse partially due to increased sphingomyelinase activity [40,77,78]. Increased sphingomyelinase activity in plasma was found in ALS patients with FUS, SOD1, and TBK1 mutations as well as transgenic mice expressing FUS-R521C; genetic inhibition of acid sphingomyelinase improved motor behavioral dysfunction in FUS-R521C mice [79]. Lipid raft disruption as well as alterations in synaptic sphingolipid metabolism can affect neurotransmitter release, endocytosis, and receptor clustering as well as axonal regeneration in NMJs [39,80,81,82,83,84]. Particularly, localization of SNAP-25 within lipid rafts can facilitate a formation of the SNARE complex contributing to the regulation of Ca^2+^-dependent exocytosis [85,86]. Synapsin is also a lipid raft resident [87] and its overexpression can promote lipid raft formation in neuronal cells, enhancing brain-derived neurotrophic factor-stimulated neurite formation [88]. Speculatively, the upregulation of SNAP-25 as well as synapsin can be a compensatory response to lipid raft disruption at the pre-onset stage. Note that lipid raft disruption and lipid peroxidation can also occur at the early stage in perisynaptic Schwann cells covering the nerve terminals. Early changes in perisynaptic Schwann cell functioning were found in the NMJs of SOD1(G37R) and SOD1G93A mice [23,89,90,91]. 

Accordingly, a strategy for the stabilization of lipid metabolism and lipid rafts can be potentially used for ALS therapy. Administration of ambroxol hydrochloride, an inhibitor of glycosphingolipid degradation in drinking water delayed disease onset, protecting NMJs, and the number of functional spinal motor neurons in SOD1G86R mice [35]. Neuron-targeted upregulation of caveolin-1, a lipid raft organizer scaffold, improved running-wheel performance, and preserved α-motor neuron morphology and NMJ integrity in hSOD1G93A mouse and rat models [36]. Upregulation of genes encoding caveolin-1 and -2 in some ALS patients can be a compensatory mechanism prolonging survival duration and slowing progression of the disease [92]. In contrast, cholesterol-lowering drug simvastatin exacerbated skeletal muscle atrophy and denervation in SOD1G93A mice [93]. Finally, mutations within CAV1/CAV2 enhancers, which reduce caveolin expression and, hence, disrupt membrane lipid rafts, are ALS risk factors [94].

## 4. Materials and Methods

### 4.1. Animal Model

Transgenic mice of the FUS1-359 genotype on a CD1 genetic background were used as the model of ALS. These mice develop the formation of aberrant FUS aggregates in the cytoplasm of motor neurons, which leads to the progressive death of the motor neurons. Transgenic ΔFUS(1-359) mice at 6–8 weeks old were used at the early presymptomatic (pre-onset) stage of the disease without any motor symptoms. Transgenic ΔFUS(1-359) mice at 18–20 weeks old were used at the symptomatic (onset) stage of the disease, with paraplegia of hind limbs. This ALS mouse model was previously characterized using histological and motor function tests [25,26,27,31,32]. Age-matched CD1 wild-type (WT) mice were used as the control group. Mice were kept in ventilated cages (3–4 per cage) under a humidity- (30–60%) and temperature- (20–24 °C) controlled environment. Animals were maintained at in 12 hr light/12 hr dark cycle and food/water was available ad libitum.

### 4.2. Microelectrode Recordings

The isolated diaphragm muscle was dissected into hemidiaphragms with phrenic nerve stubs. The nerve-hemidiaphragm preparation was pinned to the sylgard-coated bottom of the experimental chamber continuously perfused at 5 mL/min with physiological saline. Two variants of a physiological solution were used: low Ca^2+^, high Mg^2+^ (120.0 mM NaCl, 5.0 mM KCl, 0.5 mM CaCl_2_, 11.0 mM, NaHCO_3_, 1.0 mM NaHPO_4_, 5.0 mM MgCl_2_, and 11.0 mM glucose) and normal Ca^2+^ (129 mM NaCl, 5 mM KCl, 2 mM CaCl_2_, 1 mM MgSO_4_, 1 mM NaH_2_PO_4_, 20 mM NaHCO_3_, 11 mM glucose and 3 mM HEPES). The solutions were saturated with carbogen (95% O_2_ and 5% CO_2_); pH was kept at 7.3–7.4. The phrenic nerve was stimulated by suprathreshold stimuli (0.1–0.2 ms) via a suction electrode connected to an isolated stimulator (Model 2100, A-M Systems, Sequim, WA, USA). 

Electrophysiological experiments under low calcium and high magnesium in the extracellular solution allow a correct estimation of the probability and synchronicity of synaptic vesicle exocytotic events as well as real synaptic delays [43,46]. In addition, under the conditions of reduced excitability, the abnormalities of neuromuscular transmission can be expressed more clearly. Under low external Ca^2+^, presynaptic action potentials, evoked EPPs and MEPPs were recorded using a heat-polished, extracellular solution-filled extracellular microelectrode with a tip resistance of 2–3 MΩ and diameter of 2–3 μm [43,45]. At the NMJ regions, three-phase action potentials were detected. The precision of the microelectrode position was controlled during the entire experiment. The obtained signals were filtered (between 0.03 and 10 kHz) and digitized at 50 kHz. The amplitude, rise time (between 20 and 80% of the maximum amplitude), and time constants of the exponential decay of EPPs were calculated as described previously [43]. The quantal content (m) of EPPs for low eternal Ca^2+^ conditions was assessed using the equation m = N_i_/N, where N is the total number of stimuli and N_i_ is the number of successful responses [45,95]. These parameters were calculated after recording 300 EPPs in individual neuromuscular preparations. To estimate spontaneous neurosecretion, 100–150 MEPPs were recorded and amplitude–temporal parameters and MEPP frequency were calculated. 

Recording of uni-quantal EPPs under low external Ca^2+^ allows the estimation of real synaptic delays and the kinetics of neurotransmitter quantal release [42,44,45]. The synaptic delay was defined as an interval between the peak of the inward axonal sodium current and the onset of uni-quantal EPPs (point corresponding to the 20% rise phase). The time window for the synaptic delay measurement was set at 50 ms. The EPPs collected were used to plot the synaptic delay histogram with 50 µs-bin. The descending portions of the histograms were fitted to a triple exponential function: a fast decay phase (0–3 ms), intermediate decay phase (3–8 ms) and slow (8–50 ms) decay phase. These phases correspond to early synchronous, late synchronous and asynchronous synaptic vesicle exocytotic events [43,46]. To evaluate EPPs and real synaptic delays, the phrenic nerve was stimulated at low frequencies (0.5 and 10 Hz), allowing the avoidance of an appearance of multi-quantal EPPs and an intensification of spontaneous release-producing MEPPs indistinguishable from uni-quantal EPPs.

Under normal external calcium levels, end-plate currents (EPCs) and miniature EPCs (MEPCs) were registered with a standard two-electrode voltage clamp technique as described previously in detail [21,96]. The muscle fiber in the synaptic zone was impaled by two sharp borosilicate glass microelectrodes (with resistances of 4–6 MΩ and tip diameters of ~1 μm) filled with 2.5 M KCl. An interelectrode distance was ~100–200 μm. To block muscle contractions, muscle fibers were cut transversely; this procedure did not scientifically affect quantal neurotransmitter release and cable properties [21,96,97]. The membrane potential of the muscle fibers was kept at −45 mV, and the leak current was in the range of 10–20 nA. MEPCs and EPCs were digitized at 50 kHz and analyzed with Elph software 5.0 (custom made by Zakharov A.V., Kazan, Russia) [98]. Electrophysiological signals were recorded using the Axoclamp 900 A amplifier (Molecular devices, San Jose, CA, USA) and LA 2 digital I/O board (Rudnev-Shelyev, Moscow, Russia). MEPCs or EPCs were recorded from 25 to 30 different muscle fibers. In experiments with paired-pulse stimulation, the paired pulse ratio was calculated by comparing the second to first EPCs averaged over 14 trials. Quantal content (m) was assessed as the ratio of averaged EPC to the MEPC amplitude. To estimate the changes in quantal content under a single stimulus and intense stimulation, multi-quantal EPCs were induced via 0.5 Hz and prolonged 20 Hz stimulation of the phrenic nerve.

### 4.3. Immunostaining

The immunoexpression of synapsin 1, synaptosomal-associated protein 25-kD (SNAP-25) and synaptophysin were evaluated using the immunofluorescent method. The hemidiaphragms were fixed for 30 min in 3.7% paraformaldehyde in phosphate-buffered saline (PBS), rinsed in PBS for 30 min, and incubated for 30 min in 0.5% Triton-X100 in PBS and then in a blocking solution (5% donkey serum, 1% bovine serum albumin, and 5% Triton-X100 in PBS) for 20 min. After rinsing with PBS for 30 min, the samples were incubated with primary anti-synapsin-I (Abcam, Cambridge, UK, ab64581, 1:200) and anti-SNAP-25 (Abcam ab31281, 1:200) antibodies in PBS overnight at 4 °C, rinsed in PBS for 30 min and incubated with secondary antibodies (Alexa FL 488 for anti-synapsin-I antibodies, and Alexa FL 647 for anti-SNAP-25 antibodies; Life Technologies, Carlsbad, CA, USA, 1:800) and alpha-bungarotoxin conjugated with tetramethylrhodamine (αBtx, Life Technologies, 100 ng/mL) in PBS for 1 h at RT in the dark. Finally, the preparations were rinsed for 30 min at RT in the dark and mounted on glass slides with Shandon Immu-Mount. Some preparations were incubated with primary anti-synaptophysin monoclonal antibodies (ab8049, Abcam, 1:200) and then stained with secondary antibodies conjugated with Alexa FL 647 (A31571, Life Technologies, 1:800) and α-Btx conjugated with tetramethylrhodamine. Images were captured using a Leica TCS SP5 MP confocal microscope (Leica, Wetzlar, Germany). The ImageJ Fiji software (Madison, WI, USA) was used for image analysis. Each NMJ was manually identified. The fluorescence intensity of the synaptic area was determined as the average intensity of all pixels in the synaptic area minus the average intensity of all pixels in the neighboring non-synaptic region (background fluorescence).

### 4.4. Lipid Assays

Cholera toxin subunit B (CTxB) simultaneously binds to five molecules of GM1 gangliosides, and such clusters are formed within the liquid ordering phase of the plasmalemma (lipid rafts) [99]. The hemidiaphragms were incubated with CTxB conjugated to AlexaFluor 488 (1 μg/mL; Cat# C22841; Thermo Fisher Scientific, Waltham, MA, USA) for 20 min in the physiological solution and then rinsed for 30 min before image recording. The fluorescence of the marker was excited with a 488/10 nm light and emission was recorded using a 505–545 nm band-pass filter.

BODIPY FL C5-Ceramide complexed to bovine serum albumin (ThermoFisher, Cat. # B22650) was applied to estimate the incorporation of ceramide into the plasmalemma [100]. The hemidiaphragms were exposed for 1 ½ h to the fluorescent ceramide (0.1 μM) and then rinsed for 1 ½ h with the physiological solution. The fluorescence was excited with a 488/10 nm light and emission was detected using a 505–545 nm band-pass filter.

Junctional regions were defined with rhodamine-conjugated α-bungarotoxin (α-Btx;100 ng/mL; Cat. # T1175), which was added to a bathing solution together with fluorescent CTxB and BODIPY FL C5-Ceramide. α-Btx fluorescence was excited at 555/15 nm and detected with a 630/20 nm emission filter. The intensity of CTxB and BODIPY FL C5-Ceramide fluorescence was defined in arbitrary units in junctional regions and muscle fiber membranes (without NMJs).

Lipid peroxidation was assessed from a ratio of the green to red fluorescence of BODIPY 581/591 C11 reagent; an increase in this ratio indicates lipid peroxidation [96]. The hemidiaphragms were incubated with 10 μM BODIPY 581/591 C11 (Cat # D3861) for 30 min, and perfused with the normal physiological solution for 30 min. The fluorescence was recorded with filters for fluorescein isothiocyanate and Texas Red. Intensities of green and red fluorescence were measured and the ratio of green to red fluorescence was calculated. 

Fluorescence recording was performed using a BX51WI Olympus microscope with a confocal spinning disk unit, a CoolLED pE-4000 light source (CoolLED Ltd., Andover, England), water immersion objectives (UPLANSapo 60xw or LumPlanPF 100xw), DP72 (Olympus, Hamburg, Germany) and Dhyana 400BSI V2 (Tucsen, Gaishan Town, China) cameras under the control of CellSens (Olympus) and Mosaic (Tucsen) software, respectively. For dual-channel imaging, the Optosplit II bypass image splitter (Cairn Research, Faversharm, UK) was used. Images were analyzed with ImagePro (Media Cybernetics, Rockville, MD, USA).

### 4.5. Assessment of Synaptic Vesicle Endocytosis

FM1-43 (ThermoFisher Scientific, Waltham, MA, USA) styryl dye was used to estimate a compensatory endocytosis upon induction of massive synaptic vesicle exocytosis due to intense motor nerve stimulation. The dye at concentration of 6 μM was added to experimental bath 1 min before the nerve stimulation and was present during the stimulus train (50 Hz, 30s) and 3 min after the end of the tetanus. Then, the preparations were perfused in a dye-free solution for 30 min to wash out the dye bound to plasmalemma. Physiological solution with normal external Ca^2+^ was used.

Fluorescence was observed using a BX51W1 motorized microscope (Olympus, Japan) equipped with a DSU confocal scanning disk and a OrcaR2 CCD camera (Hamamatsu, Hamamatsu City, Japan) connected to a PC through an Olympus Cell^P software (Olympus, Hamburg, Germany). The optics for analyzing FM 1-43 fluorescence included a 480/30 nm excitation filter, a 505 nm dichroic mirror and a 515 nm long pass emission filter and Olympus LUMPLFL 60x (0.9 NA) water immersion objective. ImagePro Plus software version 6.0 (Media Cybernetics, Rockville, MD, USA) was used to assess the fluorescence intensity as arbitrary fluorescence units after the background fluorescence subtraction. The background fluorescence was determined as the mean fluorescence intensity in a 50 × 50 pixel square in an image area without nerve terminals [101].

### 4.6. Calcium Imaging

The changes in cytosolic Ca^2+^ levels in motor nerve terminals upon nerve stimulation were monitored as previously described [102]. Due to technical difficulties of the dye loading into nerve terminals of diaphragm, these experiments were performed on levator auris longus. Briefly, presynaptic nerve terminals were loaded via the nerve stump with a high-affinity Ca^2+^ indicator Oregon Green 488 BAPTA-1 Hexapotassium Salt (1 mM; Molecular Probes, Eugene, OR, USA) or a low-affinity Ca^2+^ probe Magnesium Green Pentopotassium salt (10 mM, Molecular Probes, Eugene, OR, USA). The former was used for estimation of [Ca^2+^]_in_ transient at 0.5 Hz stimulation, whereas the latter—to monitor the dynamics of intraterminal Ca^2+^ upon pronged nerve activity at 10, 20 and 70 Hz.

In experiments with Oregon Green 488 BAPTA-1, fluorescence was detected with an Olympus BX-51 microscope equipped with ×40 water-immersion objective (Olympus, Tokyo, Japan), Polychrome V (Till Photonics, Munich, Germany) and high-speed camera NeuroCCD-smq camera (RedShirtImaging, Decatur, GA, USA). Region of interest (80 × 80 pixels) were captured using Turbo-SM software version 4.2.1.0 (RedShirtImaging, Decatur, GA, USA) at 500 fps for detection of intraterminal calcium dynamics. Fluorescence was excited by 488 nm light and recorded with 505DCXT dichroic mirror and E520LP emission (Chroma, Bellows Falls, VT, USA). Turbo-SM software was used to record data.

In the experiments with Magnesium Green, calcium fluorescence was recorded using a laser scanning confocal microscope, Leica TCS SP5 MP, with a 20× water immersion objective (Leica Microsystems, Heidelberg, Germany) with an argon laser (488 nm) as the light source and a recording rate of 2 fps. Video images were recorded with a photomultiplier at a 512 × 512 resolution in the LAS AF software version 2.6.3 (Leica Microsystems, Heidelberg, Germany).

Images were analyzed using ImageJ software (NIH, Bethesda, MD, USA). Background fluorescence was subtracted and then relative fluorescence was calculated according to the equation (ΔF/F0 − 1) × 100%, where ΔF is the intensity of fluorescence in response to stimulation, and F0 is the intensity of fluorescence at the resting period.

### 4.7. Statistics

Statistical analysis was performed using GraphPad Prism 8 (Boston, MA, USA) and OriginPro 15 software (Northampton, MA, USA). Data are represented as mean ± standard deviation (SD); and sample size (n) is the number of individual muscles from individual mice. The sample size and number of NMJs were indicated in each figure legends. Statistical significance was assessed by two-way ANOVA with post hoc Bonferroni multiple comparisons test and Mann–Whitney *U* test. Values of *p* ˂ 0.05, *p* ˂ 0.01, *p* ˂ 0.001 were considered as significant.

## 5. Conclusions

NMJ has a high safety factor for preserving muscle activity even under the conditions of nascent pathology and decreasing neuromuscular transmission. Early alterations in structural and functional properties of NMJs can be a triggering factor for dying-back neurodegeneration in both familiar and sporadic forms of ALS. Hence, attenuations of synaptic vesicle exo–endocytosis upon intense activity accompanied by synapsin 1 upregulation, lipid raft disturbance and lipid peroxidation can be a component of the complex pathogenetic mechanism behind ALS. Then, disorganization of end plates, alterations in AChR properties as well as downregulation of presynaptic proteins and changes in timing of neurotransmitter release appear to contribute to disease manifestation. Further molecular and genetic studies directed at reversing early NMJ alterations can help to reveal the roots of FUS-dependent pathology in ALS.

## Figures and Tables

**Figure 1 ijms-24-09022-f001:**
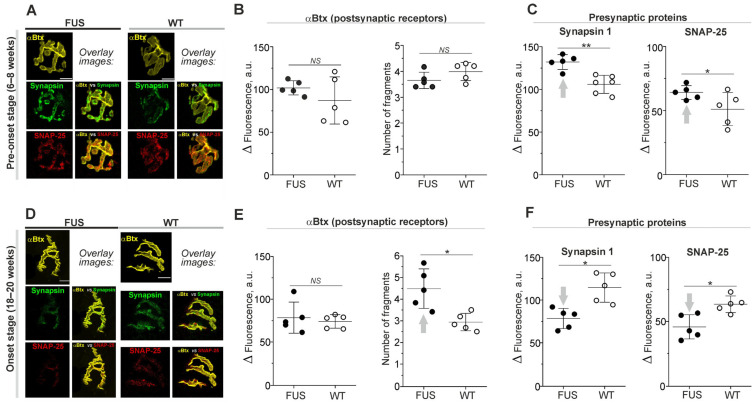
Labeling of nicotinic acetylcholine receptors (nAChRs) and presynaptic proteins in diaphragm NMJs of ΔFUS(1-359) mice vs. those of WT mice. (**A**–**F**) Comparison between ΔFUS(1-359) mice at pre-onset (**A**–**C**) and onset (**D**–**F**) stages and age-matched WT animals. (**A**,**D**) Representative fluorescent and overlay images. Scale bars—20 µm. (**B**,**E**) Fluorescence intensity of tetramethylrhodamine-conjugated αBtx and calculation of number of end-plate fragments. (**C**,**F**) Immunoexpression of synapsin 1 and SNAP-25. * *p* ˂ 0.05, ** *p* ˂ 0.01 according to Mann–Whitney *U* test compared to that for WT mice. *n* = 5–6 muscles from individual mice per group; 11–15 NMJs from each muscle. Arrows indicate an increase or decrease in parameter. *NS*—non significant.

**Figure 2 ijms-24-09022-f002:**
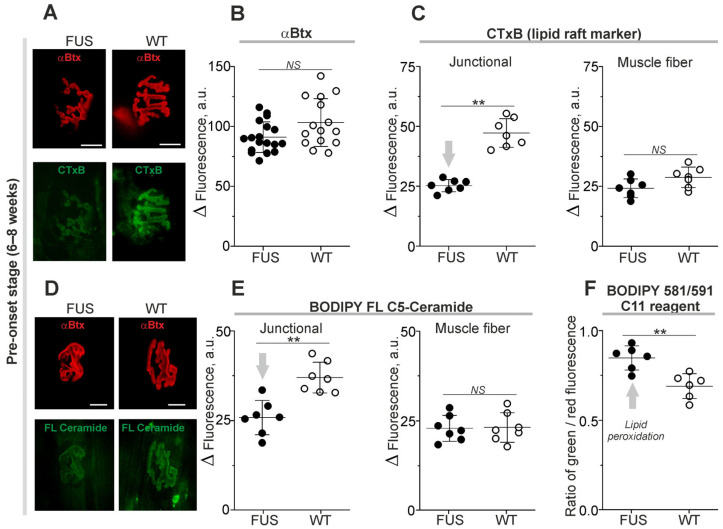
Changes in membrane properties of diaphragm muscle from ΔFUS(1-359) mice at pre-onset stage. (**A**,**D**) Typical fluorescent images of NMJs double-labeled with tetramethylrhodamine-conjugated αBtx and fluorescent CTxB or ceramide. Scale bars—20 µm. (**B**) Quantification of tetramethylrhodamine-conjugated αBtx fluorescence. (**C**,**E**) Intensities of CTxB (**C**) or ceramide (**E**) fluorescence in synaptic (junctional) and non-synaptic (muscle fiber) regions. (**F**) Ratio of green and red fluorescence of BODIPY 581/591 C11 reagent, an indication of lipid peroxidation. ** *p* ˂ 0.01 according to Mann–Whitney *U* test compared to that for WT mice. *n* = 6–7 muscles from individual mice per group (excluding 14–17 muscles for B); 14–15 NMJs from each muscle. Arrows indicate an increase or decrease in parameter. *NS*—non significant.

**Figure 3 ijms-24-09022-f003:**
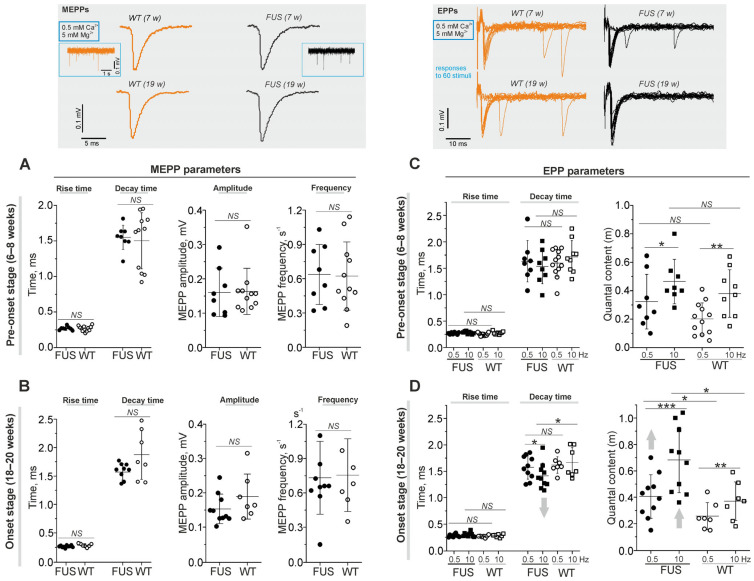
Quantal neurotransmitter release under low external Ca^2+^ in diaphragm NMJs of ΔFUS(1-359) mice vs. those of WT mice. (**A**–**D**) Comparison between ΔFUS(1-359) mice at pre-onset (**A**,**C**) and onset (**B**,**D**) stages and age-matched WT animals. Top: typical traces of MEPPs and EPPs (at 0.5 Hz stimulation) in WT vs. FUS (7- and 19-week-old) mice. In terms of EPPs, the responses to 60 nerve stimuli are shown. (**A**,**B**) MEPP parameters (rise and decay times, amplitude and frequency). (**C**,**D**) EPP characteristics: rise and decay times, and quantal contents (assessed as m = N_i_/N, where N is the total number of stimuli and N_i_ is the number of successful responses) at two frequencies of nerve stimulation (0.5 and 10 Hz). * *p* ˂ 0.05, ** *p* ˂ 0.01, and *** *p* ˂ 0.001 according to Mann–Whitney *U* test; *NS*—non significant. *n* = 6–11 muscles from individual mice per group; 100–150 MEPPs and 300 EPPs were recorded in each muscle. Arrows indicate an increase or decrease in parameter.

**Figure 4 ijms-24-09022-f004:**
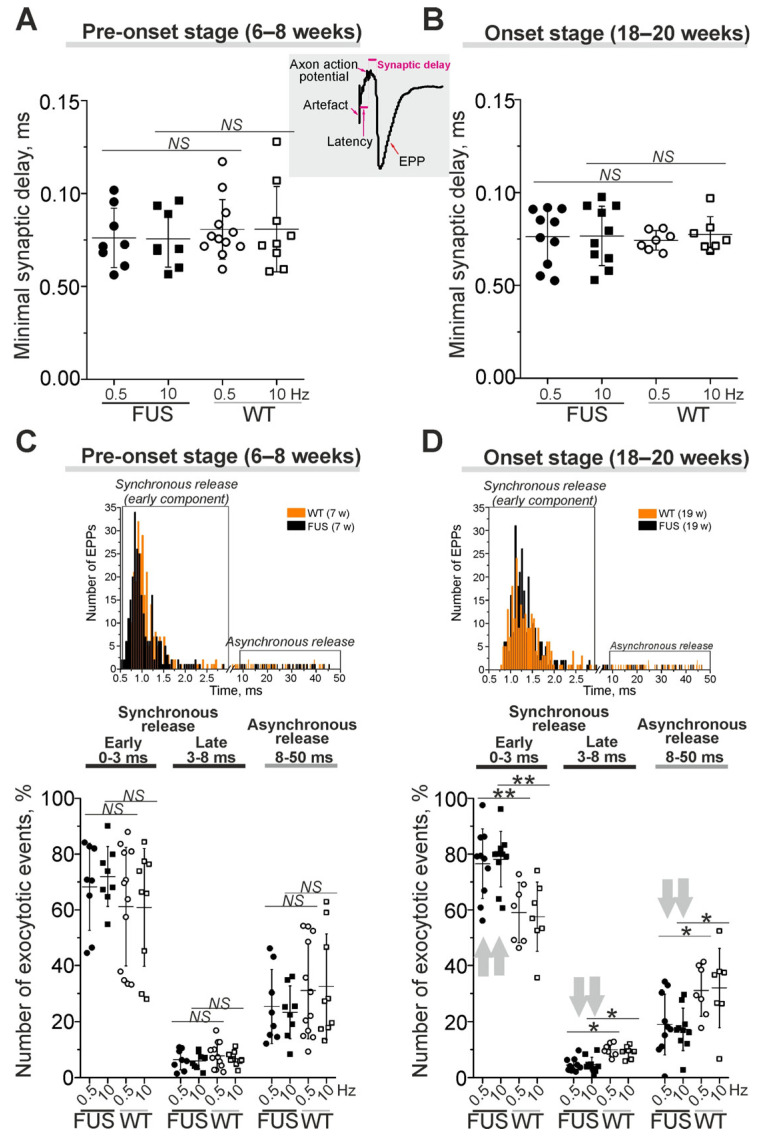
The kinetics of quantal neurotransmitter release under low external Ca^2+^ in diaphragm NMJs of ΔFUS(1-359) mice vs. those of WT mice. (**A**–**D**) Comparison between ΔFUS(1-359) mice at pre-onset (**A**,**C**) and onset (**B**,**D**) stages and age-matched WT animals. (**A**,**B**)—Quantifications of minimal synaptic delays of EPPs at 0.5 and 10 Hz nerve stimulation. Insert, synaptic delay and latency are depicted on typical EPP. (**C**,**D**)—Estimation of synchrony of exocytotic events. Shown portions (in%) of the evoked synaptic responses with short (0–3 ms), intermediate (3–8 ms) and long (8–50 ms) synaptic delays. Top, the histograms illustrating the synaptic delay distribution from separate experiments in WT and model animals; early synchronous and asynchronous components of the neurotransmitter release are boxed. * *p* ˂ 0.05, ** *p* ˂ 0.01 according to Mann–Whitney *U* test; *NS*—non significant. *n* = 7–12 muscles from individual mice per group; 300 EPPs were recorded in each muscle. Arrows indicate an increase or decrease in parameter.

**Figure 5 ijms-24-09022-f005:**
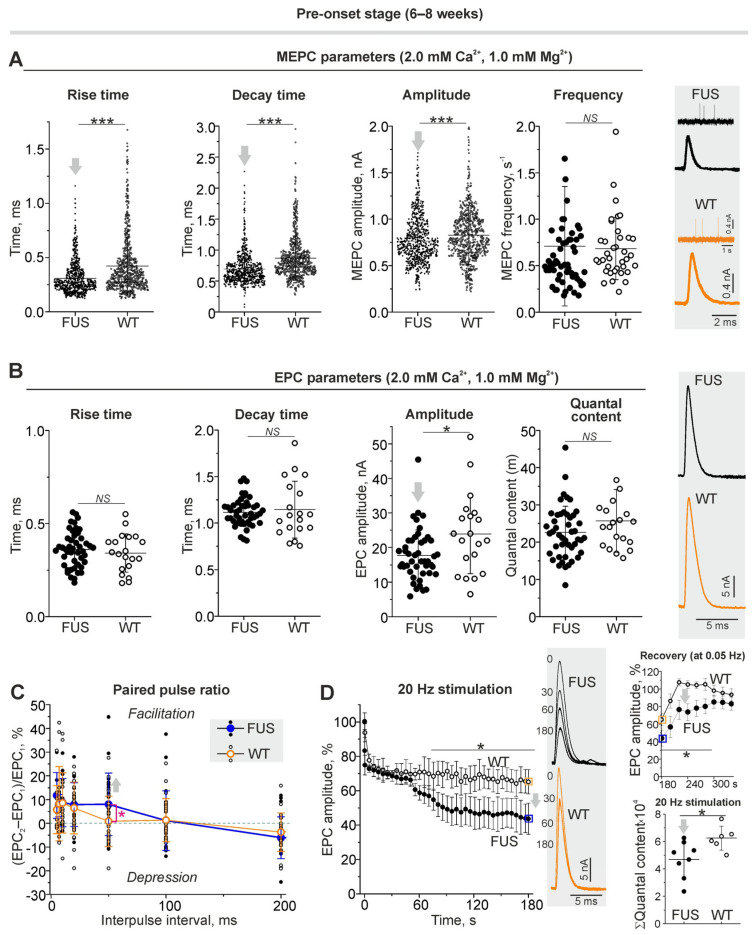
Neurotransmission under normal external Ca^2+^ level in diaphragm muscles from ΔFUS(1-359) mice at pre-onset stage. (**A**) MEPC parameters (rise and decay time, amplitude and frequency). Right: typical MEPCs. (**B**) EPC parameters: rise and decay time, amplitude and quantal content under 0.5 Hz nerve stimulation. Right: representative EPCs. (**C**) Paired pulse ratio of EPC amplitudes at inter-pulse interval from 5 to 200 ms. (**D**) Kinetics of EPC amplitude (in%) under 20 Hz stimulation for 3 min. Typical EPCs recorded at 0, 30, 60 and 180 s of stimulation are shown. Top right: recovery of EPC amplitudes (at 0.5 Hz) after 20 Hz and 3 min stimulus train; boxed points (in orange and blue) are the last values of mean EPC amplitudes at the end of tetanus. Bottom right: cumulative quantal content illustrating total amount of quanta released during 3 min stimulation at 20 Hz. (**A**–**D**) (cumulative quantal content)—* *p* ˂ 0.05 and *** *p* ˂ 0.001 according to Mann–Whitney *U* test and (**D**) according to two-way ANOVA with Bonferroni correction between curves; NS—non significant. (**A**) *n* = 5 muscles from individual mice per group; MEPCs were recorded in 140–150 NMJs in each muscle for estimation of the temporal parameters and amplitude; for MEPC frequency analysis, 200–300 MEPCs were recorded in each of the 7–12 NMJs per muscle. (**B**) *n* = 5 muscles from individual mice per group; EPCs were detected in 5–9 NMJs in each muscle. (**C**,**D**) *n* = 6–8 muscles from individual mice per group; one NMJ in each muscle was tracked. Arrows indicate an increase or decrease in parameter.

**Figure 6 ijms-24-09022-f006:**
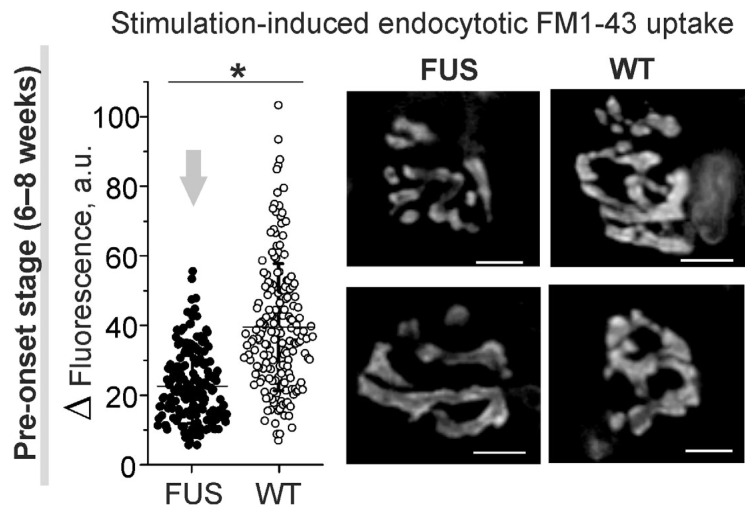
Compensatory synaptic vesicle endocytosis assessed using styryl FM1-43 dye in diaphragm NMJs of ΔFUS(1-359) mice at pre-onset stage. Left: quantification of fluorescence of individual NMJs loaded by 30 s nerve stimulation at high frequency. Right: typical fluorescent images of FM1-43-loaded motor nerve terminals. Scale bars—10 μm. *n* = 4 muscles from individual mice; 40–50 NMJs per animal. * *p* ˂ 0.05 according to Mann–Whitney *U* test between mice.

**Figure 7 ijms-24-09022-f007:**
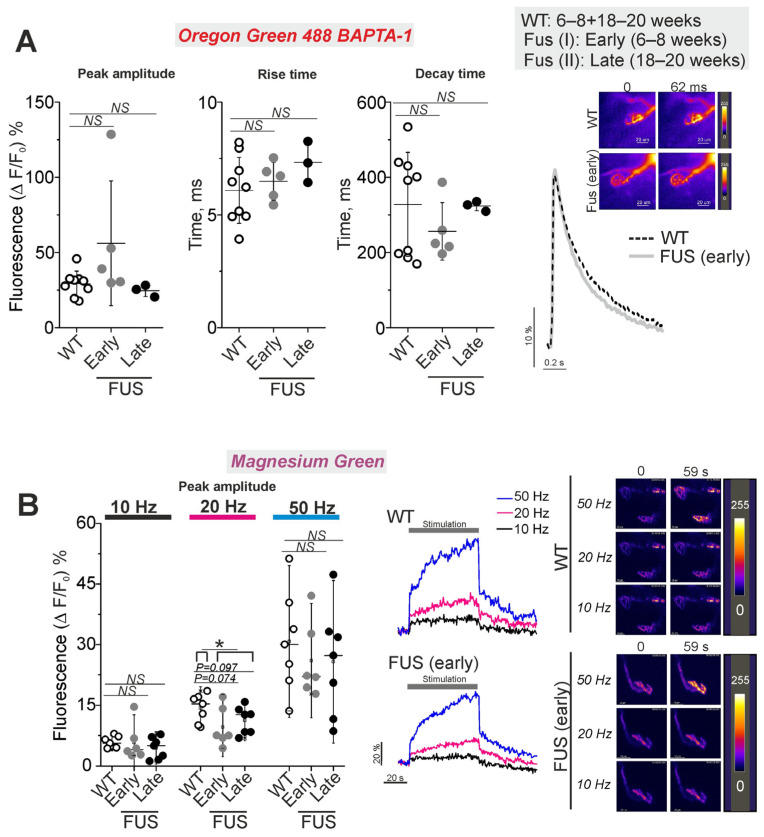
Calcium dynamics in presynaptic nerve terminals in levator auris longus muscle NMJs of ΔFUS(1-359) mice vs. those of WT mice. The figure shows a comparison between ΔFUS(1-359) mice at pre-onset (early phase, I; 6–8 week-old) and onset (late phase, II; 18–20 week-old) stages compared to WT (6–8 + 18–20 week-old) mice. (**A**) Parameters of intraterminal calcium transient upon single stimulus. WT: 47 NMJs of 9 mice; FUS I (early): 40 NMJs of 5 mice; FUS II (late): 17 NMJs of 3 mice. Right: typical traces of Ca transient and pseudocolor images of Oregon Green 488 BAPTA-1 fluorescence in the nerve terminals. Scale bars—20 µm. (**B**)Peak amplitude of one-minute stimulus train-induced increase in intraterminal calcium fluorescence of nerve terminals loaded with Magnesium Green. The motor nerve was stimulated at 10, 20 and 50 Hz. WT: 36 NMJs of 7 mice; FUS I (early): 19 NMJs of 6 mice; FUS II (late): 27 NMJs of 7 mice. Left: quantification of the peak fluorescence intensity. Center: typical traces of changes in Ca^2+^ fluorescence intensity. Right: representative pseudocolor images of Magnesium Green fluorescence in the nerve terminals. Scale bars—20 µm. * *p* ˂ 0.05 according to Mann–Whitney *U* test; *NS*—non significant. *n* = 3–9 muscles from individual mice per group; 7–10 NMJs were tested in each muscle.

**Figure 8 ijms-24-09022-f008:**
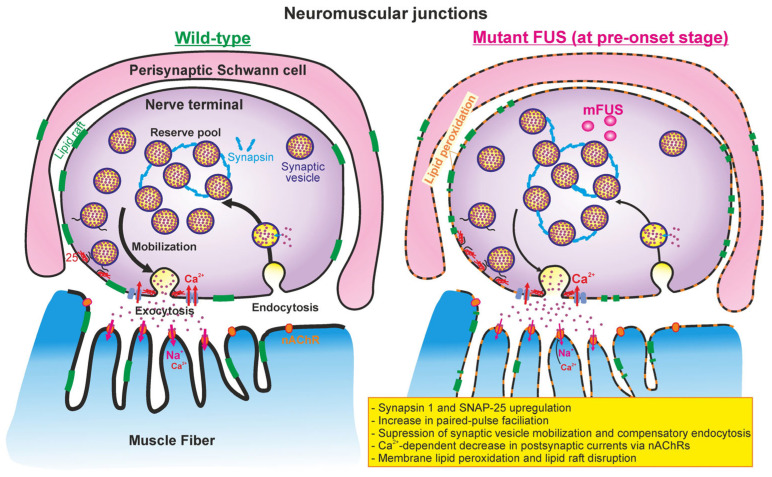
Putative model of NMJ alterations in ΔFUS(1-359) mice at pre-onset stage. (Left) presynaptic processes (calcium-induced synaptic vesicle exocytosis, mobilization, grouping into reserve pool, and endocytosis) and neurotransmitter-mediated postsynaptic currents in wild-type mice. (Right) mutant FUS-mediated changes in presynaptic and postsynaptic events in NMJs at early (presymptomatic) stage. Perisynaptic Schwann cells can also be affected at the early stage. Please see detailed explanations in the text.

**Table 1 ijms-24-09022-t001:** Morphometric analysis of labeling of nicotinic acetylcholine receptors (nAChRs) and presynaptic proteins.

Parameter	FUS (6–8 w)	WT (6–8 w)	*p*-Value	FUS (18–20 w)	WT (18–20 w)	*p*-Value
nAChR area	197 ± 20 µm	205 ± 46 µm	0.835	145 ± 29 µm	189 ± 25 µm	0.038 *
Synapsin 1 area	97 ± 24 µm	86 ± 30 µm	0.403	68 ± 17 µm	83 ± 20 µm	0.403
SNAP-25 area	80 ± 31 µm	84 ± 23 µm	0.531	65 ± 28 µm	72 ± 10 µm	1
Synaptophysin area	113 ± 12 µm	118 ± 17 µm	0.676	78 ± 9 µm	121 ± 38 µm	0.012 *
M1 coefficient *	0.51 ± 0.13	0.55 ± 0.13	0.362	0.35 ± 0.12	0.49 ± 0.13	0.067
M2 coefficient **	0.36 ± 0.09	0.32 ± 0.70	0.208	0.22 ± 0.08	0.32 ± 0.06	0.024 *

*p* ˂ 0.05 according to Mann–Whitney *U* test compared to that for WT mice; mean ± SD, *—percentage of nAChR fluorescence area covered by synapsin 1 fluorescence; **—percentage of synapsin 1 fluorescence area covered by nAChR fluorescence. *n* = 5 muscles from individual mice per group; 11–15 NMJs were calculated from each muscle.

## Data Availability

All mentioned data are represented in the main manuscript figures and Appendix A. Other additional data will be made available upon reasonable request.

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
