# Peer review of "Early Alterations in Structural and Functional Properties in the Neuromuscular Junctions of Mutant FUS Mice"

_ijms, 2023, doi:10.3390/ijms24109022_

Round 1

Reviewer 1 Report

Mukhamedyarov and collaborators performed a well-designed and elegant study to characterize early NMJ alterations in ALS-affected diaphragm. The obtained results precisely address the working objectives and add new evidence in favour of the dying back theory affecting motoneurons in ALS mice. Thus, early alterations in NMJs can be triggering factors for the disease, including disturbances as attenuation of synaptic vesicle cycle upon intense activity, synapsin 1 upregulation and lipid peroxidation that can be part of the pathogeny behind ALS. Also, end plate disorganization, alterations in AChR properties, presynaptic proteins downregulation and changes in neurotransmitter release can appear contributing to disease manifestation.

There are only a few details that authors could address to improve the quality of the manuscript:

-          Would suggest moving lines 153-158 to M&M section.

-          The authors could explain why they chose to evaluate EPPs at 0,5 and 10 Hz and why 7. calcium dynamics in presynaptic nerve terminals were evaluated at 10, 20 and 50 Hz in the M&M section.

-          Line 210: “the” should be removed

-          In Figure 4 C and D it should be indicated whether each graph concerns to presymptomatic and onset stage.

The quality of English is good. 

Author Response

Response to Review #1

General response: We are grateful to the Reviewer for positive evaluation of the work and helpful comments. In the revised version, all raised questions were addressed to improve quality of the manuscript.

Response to specific comments:

1) We agree. The text (lines 154-158) was moved to M&M section as recommended.

2) To clarify this issue, the corresponding explanations were added in the M&M section.

Both microelectrode recordings (at low and normal external Ca2+) and Ca2+ imaging were performed at 0.5 Hz stimulation (Fig. 3C, D, Fig. 5B, & Fig 7A). Then Ca2+ dynamics were tested at higher frequencies (10, 20 and 70 Hz), where a trend to decrease in the intarterminal calcium has been found upon 20 Hz activity (Fig. 7B; Suppl. Fig. 2). Accordingly, 20 Hz prolonged stimulation was chosen for electrophysiological experiments at normal external Ca2+ (Fig. 5D). At low external Ca2+ experiments, 0.5 and 10 Hz stimulations were only used (Fig, 3C, D & Fig. 4), since an increase in frequency of stimulation under these conditions makes it difficult to assess probability of exocytotic events and real synaptic delay due to an appearance of multi-quantal EPPs and intensification of spontaneous release producing MEPPs indistinguishable from uni-quantal EPPs.

In 4.2, low external calcium conditions: “To evaluate EPPs and real synaptic delays, the phrenic nerve was stimulated at low frequencies (0.5 and 10 Hz) allowing to avoid an appearance of multi-quantal EPPs and intensification of spontaneous release producing MEPPs indistinguishable from uni-quantal EPPs.”

In 4.2, normal external calcium conditions: “To estimate the changes in quantal content upon single stimuli and intense stimulation, multi-quantal EPCs were evoked by 0.5 Hz and prolonged 20 Hz stimulation of the phrenic nerve”

In 4.8, calcium imaging: “Briefly, presynaptic nerve terminals were loaded via the nerve stump with a high-affinity Ca2+ indicator Oregon Green 488 BAPTA-1 Hexapotassium Salt (1 mM; Molecular Probes, Eugene, OR, USA) or a low-affinity Ca2+ probe Magnesium green Pentopotassium salt (10 mM, Molecular Probes, Eugene, OR, USA). The former was used for estimation of [Ca2+]in transient at 0.5 Hz stimulation, whereas the latter – to monitor the dynamics of intraterminal Ca2+ upon pronged nerve activity at 10, 20 and 70 Hz.”

3) This grammar issue (line 210) was corrected.

4) Indications of pre-onset and onset stages were added in Fig 4 C & D.

5) The gramma was checked.

Reviewer 2 Report

The study by Marat A. Mukhamedyarov et al. focuses on early structural and functional alterations of diaphragm NMJs for FUS mutation mice at the pre-onset stage. However, there are a few concerns that the authors need to address:

Major points:

1.      The author needs to draw a model to make a clear to show how FUS mutation alters the NMJ and its transmission.

2.      There are no NMJ function tests for FUS mutation mice such as muscle strength, and rotarod test.

3.      In Figures 1-7, the author should label all Y axis. Figure 1C should be put on the first panel (Figure 1A).

4.      The author should write clearly which muscle is used for recording. For the mEPP should show at least 5-10s trancing imaging and amplitudes so small. In addition, the EPP amplitude is so small. It looks unnormal for these results.

Minor points:

5.      In Figure 2. Fus and WT should be put out of the panel for X axis. All should be consistent.

6.      The author should explain which muscle is used for staining such as TA, GS or EDL muscle, and write it clearly in your paper.

7.      The author should explain why wants to test lipid raft and ceramide in the NMJ for Fus and WT mice. List the original paper to support it.

8.      The author needs to record CMAP in the early stage for Fus and WT based on the Fus mutation mice increased NMJ fragments. 

Author Response

Response to Review #2

General response: We are grateful to the Reviewer for evaluation of the work and helpful comments on the manuscript. In the revised version, we addressed the raised questions and added additional explanations.

Response to specific comments:

1) The proposed model was added as figure 8 (in the Discussion) and graphical abstract.

2)  This model was characterized in terms of functional (motor) test (indulging rotarod, muscle strength, wire, pole tests and gait analysis) previously. We added information and references (doi:10.1074/jbc.M113.492017, doi:10.1007/s10048-018-0553-9, doi:10.1080/21678421.2017.1407796, doi:10.1111/cns.13280, doi:10.1016/j.biopha.2022.113986) in 4.1 section of M&M. At 6-8 weeks (pre-onset stage) these tests did not reveal any deviations from control values.

Herein, we focused on pre-onset stage. In some experiments with symptomatic mice (18-20 weeks), the animals with apparent paraplegia of hind limbs were taken in the experiments (this was noted in 4.1 section). This is consistent with literature data where motor tests revealed clear defects at 18-20 weeks.

We also have data from wire-test (there is no difference at 12 weeks, but the decline occurred at 16 weeks). We are planning these results to another paper aimed to efficacy of a treatment of the model mice. 

3) We agree. The figures were corrected. Fig. 1 was rearranged.

4) All experiments (excluding Ca2+ imaging due to technical reasons) were carried out on diaphragm muscle. This is indicated in the introduction (last paragraph), discussion and in M&M section. As it was recommended, we added the indication of muscle in each section.

We added 5-s recording of MEPPs on Fig. 3 as well as Fig. 5. In cases of Fig. 3 & 4 the MEPPs and EPPs were “small” since they were recorded using extracellular electrodes at low calcium (0.5 mM) and high magnesium (5 mM) in extracellular solution. Under these conditions, uni-quantal EPPs are generated upon the nerve stimuli. Accordingly, these values of the amplitudes are normal for this type of microelectrode recording. This information is described in 4.2 section of M&M.

We also recorded MEPCs and EPCs at 1.8 mM Ca2+ and 1 mM Mg2+ in extracellular solution (Fig 5). Indeed, under these conditions the amplitudes of MEPCs and especially multi-quantal EPCs are much higher.

To draw the reader's attention to low external calcium in the experiments of section 2.3, we added the notion in text.  Section 2.3.1: “Postsynaptic electrical responses were recorded using extracellular microelectrode at 0.5 mM Ca2+ and 5 mM Mg2+ in physiological solution [29-31].”

5) Figure 2 was corrected to be consistent.

6) In each result section and figure legend, we added the information of muscle used.

7) The justification of tests for lipid rafts and ceramide was added. The full list of original papers revealing an involvement of lipid rafts and ceramide in ALS are in the Discussion (please see subsection “The alterations of synaptic lipids in NMJs of FUS mice

In 2.2 section: “ Previous studies on mutant superoxide dismutase (SOD1) mice revealed changes in lipid rafts and signs of ceramide metabolism dysregulation in both spinal cord and NMJs [21,25-27]. Furthermore, increased expression of lipid raft scaffold protein caveo-lin-1 as well as inhibition of glucocerebrosidase GBA2 (an enzyme involved in ceramide production) were beneficial in mSOD animals [27-29].” ).

In addition, we added the description of two recent studies on lipid raft-ALS link in the Discussion. 

8) We agree that CMAP recording is helpful technique to such type of studies, but we do not have the opportunity to do these experiments. Note that the increased number of NMJ fragments was observed only at onset (but not pre-onset stage).

Reviewer 3 Report

In this manuscript, Marat and colleagues investigated alterations in diaphragm neuromuscular junctions in an ALS mouse model. They performed an extensive characterization of the diaphragm neuromuscular junctions (NMJs) in ΔFUS(1-359) mice both at pre-onset and onset stage. The authors use a range of techniques to study the structural and functional properties of those neuromuscular junctions: immuno- and lipid-staining, ion imaging and electrophysiology. They identified structural and functional changes in pre-symptomatic stage (6-8 weeks of age) that they foresee has potential new avenues to design strategies against fast degeneration of neuromuscular junctions in ALS, a current fatal neurodegenerative disease.  

Indeed, this study could provide additional insights into our understanding of the onset of ALS at early stages. The experiments in the manuscript are well designed and executed.  Nevertheless, it is often quite hard to link findings between different sets of experiments. Moreover, the current version of the manuscript lacks further functional validation and discussion of their findings in the bigger context. Additionally, we strongly recommend reorganizing the order of Figures and to propose a clear molecular model. This manuscript fits the scope of this journal and should be considered for publication after a few major concerns and some minor concerns have been addressed.

Major concerns:

Data

1-    Figure 1: The number of imaged NMJs is relatively small, usually the number of NMJs used for analysis would be 3-7 times higher (250-500 NMJs per group). We recommend to either increase the number of NMJs or to plot the nAChR area size in a graph to be able to judge the variability within and between samples.

2-    Figure 1: The reduction in presynaptic proteins (Synapsin1, SNAP-25) in a symptomatic stage could be due to denervation. We would recommend normalizing the surface expression of these proteins to presynaptic markers, such as SV2, VAChT or Synaptophsyin, in combination with Neurofilament which are regularly used to quantify the innervation state of NMJs.

3-    Figure 2: The authors have an interesting finding regarding the early alteration of the lipid composition of presynaptic membrane in neuromuscular junctions of mutant FUS mice. These effects seem to be quite profound. However, they did not investigate whether these alterations persist as the disease progresses. Since in Figure 1, the expression pattern of Synapsin-1 and SNAP25 revert in later stages of the disease, it appears to be a very important assessment.

4-    There is a puzzling aspect in this paper by Marat and al., the early structural changes they measure are presynaptic (protein level alteration and membrane lipid composition) while the functional alterations they measure appear to be postsynaptic. 

5-    Currently, the authors fail to make a connection between the increase in Synapsin-1 and SNAP25 protein and the alteration of the membrane lipid composition/lipid raft organization. The authors should try to address this gap with additional experiments and at the very least propose a model.

6-    Figure 3: the current display of raw EPPs data do not allow the reader to assess visually the effects measured on the quantal content at 18-20 weeks. However, looking at the raw traces it seems the mutant FUS at early and later stage might decrease EPPs failure rate. Have the authors investigated this parameter? This could relay to findings in Figure 4. The phenomenon seems to appear commitment: over-priming or increase release probability of the ready to be release pool and a deficiency in recycling. The authors should discuss this further in the light of their structural findings.

7-    Figures 5, 6 and 7: The authors should describe in more details their model/hypothesis of how extracellular calcium concentration impact nAchR postsynaptic abundance of biophysical properties etc. The link is very fussy although it is central to this study.

8-    I suggest reorganizing the order of Figures -for instance move Figures 5 and 6 after Figure 2. In fact, the figures are described in a different order than the results and discussion sections. Also, data in Figure 3 (low calcium concentration) could be moved to supplemental.

9-    Lines 295 -297: How do the authors come to the conclusion that mutant FUS have more synapses?

10- As mentioned above, a discussion about the bigger picture is missing. How are the observed changes are connected? What is the hypothesis on disease onset and progression when taking into account these data? Please provide propose a model.

Additional literature.

11- The authors should discuss the choice of the mouse model and advantages/disadvantages of this model in the introduction. Furthermore, they should describe how the findings can be connected to the loss of function and gain of toxic function of ΔFUS(1-359).

12- The authors should provide references justifying the choice of 6-8 weeks and 18-20 weeks for the asymptomatic and symptomatic phase of mutant FUS ALS in mice respectively.

13- The authors should compare the functional characterization of NMJs in ΔFUS(1-359) mice to the NMJ characterization of other ALS mouse models, e.g. to the findings of Rocha, PLoS One, 2013.

Minor concerns:

1-    Line 49: ...what seems to BE A crucial even in …

2-    Line 161: spell out miniature end-plate potential (MEPP) prior to using the abbreviation.

3-    Line 210: … , was THE similar in FUS and WT…

4-    Line 324: Please indicate exact p-value as shown in Suppl. Figure 2.

5-    Line 368:…, latter the…

6-    Lines 416-426: Authors discuss an interesting hypothesis, that could be checked: are there changes in ATP and mitochondria in these mice in the diaphragm NMJs?

7-    Figure 2: Replace green/red fluorescence by the protein names.

8-    Figure 7: Nomenclature FUS I/II should be early and late to match the rest of the manuscript.

9-    Legend Figure 7B: Please explain left, center and right panel of this figure.

Author Response

General response:

We thank the Reviewer for helpful critics and comments on the manuscript. According to the concerns, new data (synaptophysin labeling, graph illustrating a distribution of nAChR area values), information and proposed model of early NMJ alterations were added in the revised version. We also want to keep the focus of manuscript on characterization of the early events (both structural and functional) in this FUS model mice and comparison with other mutant FUS models. We believe that the mechanistic insights and interconnections of observed early alterations into one clear picture of pathogenesis are topics for future separate studies. Of course, we understand that mechanistic study is much more valuable and interesting, but such extensive descriptive work could serve as starting point for multiple mechanistic investigations.

Response to major comments:

1) These data were obtained side by side for a precise comparison of groups using Leica TCS SP5 MP confocal microscope. Only surface NMJs, where all elements lie in one focus, were analyzed. The area ± SD (where n is number of animals) are in table 1; and SD is only 10-20% indicating acceptable data variability. Also, in Suppl figure 1A the area measurements of individual NMJs were provided. These estimations (Suppl Fig. 1A) gave the same results as in Table1. From statistical point of view, our estimations fully meet requirements.

2) Thank you for this comment. We agree that denervation can take place at this stage and contribute to the downregulation of the presynaptic proteins. Quantification of synaptophysin fluorescence (area and intensity) were provided in Table 1 and Suppl Fig. 1B, C. Decrease in both area and intensity of the staining was observed selectively at onset stage.

In 2.1 section: “Hence, the signs of presynaptic and postsynaptic disorganization probably due to the beginning of denervation appear in the diaphragm muscle at disease manifestation stage.”

In the Discussion: “…end plate shrinking and fragmentation, decrease in synapsin 1, SNAP-25 and synaptophysin levels, synchronization of exocytotic events occurred in the NMJs of ΔFUS(1-359) mice at onset stage. Probably, these late alterations could be consequences of beginning of the muscle denervation.”

3) We agree that it is interesting direction to track the progression of membrane changes in mutant FUS mice, but we have no ability to do this for current work. We are planning this section for separate study with evaluation of treatment effect with inhibitors of lipid peroxidation. In the current study, we have been focused on pre-onset alterations (as capitalized in abstract, introduction and conclusion).

4) As indicated in abstract, discussion and conclusion, functional changes are both postsynaptic and presynaptic. The former alterations were revealed due to recording MEPPs/MEPCs at low/normal external calcium, the latter – when EPCs during paired pulse stimulation, high-frequency activity and recovery after tetanus were analyzed. Also, decrease in endocytotic dye uptake points to functional presynaptic changes.

5) We agree that there is now experimental evidence about connection between upregulation of synapsin-1 / SNAP25 and the alteration of the membrane lipid composition/lipid raft organization. These phenomenona can be independent or dependent in very complex manner. Hence, separate studies can reveal or show independence of protein and membrane changes. To address this comment, we supplemented the Discussion with data which can help to imagine a possible interaction.

In the Discussion (in subsection: the alterations of synaptic lipids in NMJs of FUS mice): “…Lipid raft disruption as well as alterations in synaptic sphingolipid metabolism can affect neurotransmitter release, endocytosis, receptor clustering as well as axonal regeneration in NMJs [39,80-84]. Particularly, localization of SNAP-25 within lipid rafts can facilitate a formation of SNARE complex contributing to regulation of Ca2+-dependent exocytosis [85,86]. Synapsin is also lipid raft resident [87] and its overexpression can promote lipid raft formation in neuronal cells, enhancing brain-derived neurotrophic factor-stimulated neurite formation [88]. Speculatively, the upregulation of SNAP-25 as well as synapsin can be compensatory response to lipid raft disruption at the pre-onset stage.”

…“Neuron-targeted upregulation of caveolin-1, lipid raft organizer scaffold, improved running-wheel performance, preserved α-motor neuron morphology and NMJ integrity in hSOD1G93A mouse and rat models [36]. Upregulation of genes encoding caveolin-1 and 2 in some ALS patients can be a compensatory mechanism prolonging survival duration and slowing progression of the disease [89]. In contrast, cholesterol-lowering drug simvastatin exacerbated skeletal muscle atrophy and denervation in SOD1G93A mice [90]. Finally, mutations within CAV1/CAV2 enhancers, which reduce caveolin expression and, hence, disrupt membrane lipid rafts, are ALS risk factors [91].”

6) Fig 3C,D shows uni-quantal EPPs recorded at low external Ca2+. It is correctly that the failure rate of EPPs was statistically significant decrease at onset stage. This is what raw EPPs traces were shown. Under these conditions, quantal content (m) is estimated as the ratio of number of stimuli accompanied by EPP to the total number of stimuli [42,44,45]. This value is inverse to failure rate of EPPs.

We highlighted this in the results and method sections. Also, the discussion was extended.  

In 2.3.1. section: “At low [Ca2+]out, presynaptic action potential causes uni-quantal EPP and quantal content (m) is assessed as the ratio of number of stimuli accompanied by EPP to the total number of stimuli [42,44,45]. This ratio (m) correlates inversely with failure rate of EPPs.

“In 4.2 (methods section): “The quantal content (m) of EPPs for low eternal Ca2+ conditions was assessed from the equation: m=Ni/N, where N is the total number of stimuli and Ni is the number of successful responses [45,92]. These parameters were calculated after recording 300 EPPs in individual neuromuscular preparation.”

In Discussion: “Only at later stage, at low [Ca2+]out quantal content as well as portion of synchronous exocytotic events were elevated despite decreased expression of SNAP-25 and synaptophysin at the NMJs of ΔFUS(1-359) mice. Such synchronization of neurotransmitter quanta release can be a compensatory reaction to alleviate failure of neuro-transmission [46]. One possibility is that decreased synapsin expression can synchronize neurotransmitter release and increase the release probability or size of ready releasable pool [61-63].”

7) We think that this is important fact along with other presynaptic changes. The discussion was extended.

“Interestingly, that the amplitude, rise and decay times of miniature end plate re-sponses were slightly decreased at normal (but not low) external Ca2+ level in NMJs of ΔFUS(1-359) mice at pre-onset stage. This suggests a Ca2+-dependent partial inactivation of postsynaptic nAChRs as a potential early event in the ALS model. In this scenario, the influx of Ca2+ via nAChRs can negatively modulate their activity or external Ca2+ can act on nAChRs indirectly, e.g. via Ca2+-sensing receptors [48,53]. Accordingly, a decrease in external Ca2+ concentration could remove the inhibition of nAChRs in ΔFUS(1-359) mice. Note that Ca2+ influx through nAChR can activate inositol 1,4,5-trisphosphate receptors enriched in postsynaptic compartment surrounding the subsynaptic nuclei, and this calcium signaling is crucial to the stabilization of the NMJs [48,54]. Hence, the sup-pression of currents via nAChRs at the pre-onset stage might be a reason of NMJ dis-organization in ΔFUS(1-359) mice at the later stage…”.

8) We understand rationality of such reorganization of the manuscript, but it will be helpful in case of mechanistic study. The current research is descriptive with a simple logic: structural changes (proteins and lipid), functional changes (at low calcium and normal calcium). So, we kindly ask the Reviewer to allow us to keep this order of Figures. 

9) This phrase was corrected. “Furthermore, analysis of FM1-43 fluorescence intensity in individual NMJs revealed an increase in portion of nerve terminals with reduced FM 1-43 loading in FUS mice vs WT mice.”

We do not conclude that mFUS mice have more synapses in the diaphragm, we just describe that portion of nerve terminals characterized by low FM1-43 endocytotic uptake was more in the mFUS mice (please see distribution of values in Fig. 6).

10) The putative model was provided (Fig. 8). At the same time, the accumulated data do not yet allow creating an acceptable model for describing the development of the disease progression based on early signs. Hence, we discussed the postsynaptic, presynaptic and lipid abnormalities in separate sections of discussion.

Additional literature.

11) Additional information about this model was added.

In the Introduction: “These model mice express human truncated FUS 1–359 protein which lacks RNA binding capacity and is predominantly cytoplasmic and highly aggregate-prone [25]. This model allows mainly to reveal the pathological role of FUS aggregation rather than dysregulation of FUS RNA targets due to expression of mutant full-length FUS [25,26]. ΔFUS(1-359) mice recapitulate key hallmarks of ALS including oxidative stress and elevated levels of pro-inflammatory cytokines in spinal cord, FUS-positive inclusions in motor neurons, microglial activation, limb paralysis, muscle wasting and denervation, loss of spinal motor neurons, abrupt disease onset and fast progression [27-31].”

Where it was possible, we discussed the putative link between mutant FUS and observed changes.   

e.g. “. These postsynaptic defects are attributable to toxicity of the cytoplasmic FUS protein in muscle fibers and the decreased ability of normal FUS located in subsynaptic nuclei to increase expression of genes (e.g., nAChR subunits) important for NMJ stability [6]”.

….“ and FUS accumulation in early disease stage can lead to stabilization of synaptic RNA, probably contributing to synaptopathy [55]”

…” …FUS participates in anchoring of mitochondria into motor nerve terminals via interaction with syntaphilin, and FUS mutations affect mitochondria number and mobility in the synapses [14]”

12) This information was added in the Introduction.

“Transgenic ΔFUS(1-359) mice have no pathological changes and any motor symptoms until 10 weeks of age (pre-onset stage) and fast disease progression occurs during 10-18 weeks of age with severe motor phenotype [25-27,31,32]”.

13) We extended the Discussion, including comparison with the findings of Dr Rocha et colleagues. But we want to be focused on evidence from FUS mice models. The comparison of findings from many ALS models is an excellent topic for review paper (in the current form the paper already contains 99 ref).

In the Discussion: “Another postsynaptic changes in form of the increase in the mean amplitude of miniature end plate responses and decrease in their rise time were found in the NMJs of SOD1(G93A) mice at presymptomatic phases of the disease [20,21]. This suggests that partially different alterations can occur in various ALS models.”

“Noticeably, that an increase in the quantal content of EPPs upon low frequency simulation was observed at the pre-symptomatic in high copy number (SOD1G93A)1Gur strain [20], but not low-expressing (SOD1-G93A)dl1Gur strain [21].”

Response to Minor concerns:

1) It was corrected.

2) It was corrected.

3) It was corrected.

4) Exact P was added.

5) It was corrected.

6) We added in the Discussion: „In ΔFUS(1-359) mice, ATP production and mitochondria functioning have not been tested yet.”

7) There are no protein names. This is “the ratio of green and red fluorescence of BODIPY 581/591 C11 reagent”.

8) It was corrected.

9) These details were explained.

Round 2

Reviewer 2 Report

The author addressed all my concerns. But, there is still a question that needs to be solved.

1. In Figure 8, the author missed the terminal Shawan cell in the NMJ model. The author should add it in Figure 8. 

Author Response

We are grateful the Reviewer for this suggestion. We added pSC on Fig. 8 as well as notion (in the Discussion) about possible changes in pSC at the early stage. 

 "Note that the lipid raft disruption and lipid peroxidation can also occur at the early stage in perisynaptic Schwann cells covering the nerve terminals. Early changes in perisynaptic Schwann cell functioning were found in NMJs of SOD1(G37R) and SOD1G93A mice [23,89-91]."